# Causal discovery from observational and interventional data across multiple environments

**Adam Li**
Department of Computer Science, Columbia University
adam.li@columbia.edu

**Amin Jaber**
Synlico Inc.
amin.jaber@synlico.com

**Elias Bareinboim**
Department of Computer Science, Columbia University
eb@cs.columbia.edu

## Abstract

A fundamental problem in many sciences is the learning of causal structure under-lying a system, typically through observation and experimentation. Commonly, one even collects data across multiple domains, such as gene sequencing from different labs, or neural recordings from different species. Although there exist methods for learning the equivalence class of causal diagrams from observational and experimental data, they are meant to operate in a single domain. In this paper, we develop a fundamental approach to structure learning in non-Markovian systems (i.e. when there exist latent confounders) leveraging observational and interventional data collected from multiple domains. Specifically, we start by showing that learning from observational data in multiple domains is equivalent to learning from interventional data with unknown targets in a single domain. But there are also subtleties when considering observational and experimental data. Using causal invariances derived from do-calculus, we define a property called S-Markov that connects interventional distributions from multiple-domains to graphical criteria on a selection diagram. Leveraging the S-Markov property, we introduce a new constraint-based causal discovery algorithm, S-FCI, that can learn from observational and interventional data from different domains. We prove that the algorithm is sound and subsumes existing constraint-based causal discovery algorithms.

## 1   Introduction

Causal discovery is the process of learning cause-and-effect relationships between variables in a given system, which is many times the final goal of the data scientist or a necessary step towards a more refined causal analysis [1, 2]. The learning process typically leverages constraints from data to infer the corresponding causal diagram. However, it is common that the data constraints do not uniquely identify the full diagram. Therefore, the target of analysis is often an equivalence class (EC) of causal diagrams that encodes constraints found in the data (implied by the underlying unknown causal system).

An EC encodes invariances in the form of graphical constraints, and thus is used to represent all causal diagrams that encode those constraints and invariances. Formal characterizations of ECs are

37th Conference on Neural Information Processing Systems (NeurIPS 2023).

important to understand the output of a learning algorithm and how it relates to the underlying causal system the scientist aims to explain.

ECs are defined with respect to distributional invariances which are implied by the structure of the graph. For example, conditional independences (CI) are implied by d-separations in the causal graph. Hence, it is desirable to formally characterize the EC in the general setting where we have interventional data from multiple domains. A complete graphical characterization would enable i) an efficient representation of the distributional invariances in the data and ii) the ability to translate these data-invariances to graphical constraints (e.g. d-separation).

An early example of an EC when only observational data is available in a single domain is the Markov equivalence class (MEC). The MEC characterizes causal diagrams with the same set of d-separation statements over observed nodes [2–5]. Given interventional (i.e. experimental) data, one can reduce the size of the equivalence class [6, 7]. In the case of known intervention targets, the EC is known as the $\mathcal{I}$-MEC [7–9] and in the case of unknown targets, it is called the $\Psi$-MEC [6].

In prior research, domain-changes and interventions were treated similarly [10–14]. Nevertheless, various examples across scientific disciplines highlight their distinction (see Table 1). For instance, when extrapolating data-driven conclusions from bonobos to humans, consider Figure 1(b). Notably, the environment/domain, represented by the S-node pointing to $X$, illustrates differences in kidney function between the species. When applying a CRISPR intervention to a gene linked to kidney protein production ($X$), researchers investigate the impact of medication ($Y$) on fluid balance in the body ($Z$). This intervention is explicitly different from the kidney-function differences between bonobos and humans because the change-in-domain is there regardless of whether or not an intervention is made. This differentiation between interventions and domains holds significance, especially in causal discovery. By leveraging invariances across observational and interventional data from both bonobos and humans, one can learn additional causal relationships. Moreover, conflating these qualitatively distinct settings is generally invalid, as pointed out in transportability analysis [15]. Pearl and Bareinboim (2011) introduced clear semantics for S-nodes (environments), offering a unified representation in the form of a selection diagram.

In this paper, we investigate structure learning when mixtures of observational and interventional data (known and unknown targets) across multiple domains are available. The multi-domain setting has been analyzed from the lens of selection diagrams, where selection nodes (or S-nodes) encode distributional changes in the mechanisms, or exogenous variables due to a change in domain [16–18]. We will show in this paper a characterization of the EC for selection diagrams. Generalizing the structure learning setting to multiple domains requires a formal treatment because it is a common scenario in the sciences [19–31]; see Table 1 for an example of different settings and related literature). For example, in single-cell sequencing analysis, scientists are interested in analyzing the causal effects of proteins on one another. However, they may typically collect observational and/or experimental data from multiple labs (i.e. domains) and wish to combine them into one dataset. Also, scientists may collect observational and experimental data over multiple species in order to learn more about one specific species, or the relationships among species [25, 27, 32].

The celebrated FCI algorithm and its variants learn a partial ancestral graph (PAG), an MEC of causal diagrams, given purely observational data [1, 2, 33]. The $\mathcal{I}$-FCI (with known targets) and $\Psi$-FCI (with unknown targets) generalize these results to interventional data, and further reduce the size of the EC to an $\mathcal{I}$-PAG and $\Psi$-PAG, respectively [6, 7]. However, these algorithms operate in a single domain, or environment and do not account for combining known/unknown target interventions.

Various approaches have been proposed throughout the literature for causal discovery from multiple domains. The works in [10, 13, 34–38] assume Markovianity, a functional model (e.g. linearity) holds, and/or do not take into account arbitrary combinations of observational and interventional data with known and unknown targets. Alternatively, JCI pools data together and performs learning on the combined dataset [14]. Pooling data is an incomplete procedure when considering interventional data within a single domain let alone multiple domains [6][Appendix D.2].

In this paper, we take a principled approach to the multi-domain structure learning problem and formally characterize S-PAGs, the object of learning. This paper introduces the selection-diagram FCI algorithm (S-FCI) that learns from a mixture of observational and interventional data from multiple domains to construct an EC of selection diagrams, an S-PAG. Specifically, we contribute the following:

| Domain | Obs. | Interv. $\mathcal{K}$ | Interv. $\mathcal{U}$ | Property | FCI-variant | Related Lit. |
|---|---|---|---|---|---|---|
| 1 | ✓ | x | x | Markov [39] | [2, 33, 40–43] | [30, 31] |
| 1 | ✓ | ✓ | x | I-Markov [7, 44] | [7, 8, 44] | [22, 30] |
| 1 | ✓ | x | ✓ | $\Psi$-Markov [6] | [6, 13, 45] | [22, 27, 46] |
| k | ✓ | x | x | $\Psi$-Markov (Thm. 1) | [6] (Cor. 5) | [20, 21, 23, 24, 47, 48] |
| k | ✓ | ✓ | ✓ | S-Markov (Thm. 2) | S-FCI (Thm. 3) | [20–25, 28–31, 46–50] |

Table 1: Summary of Markov property results, and related algorithms that learn the ancestral graph based on number of domains and types of interventional (interv.) data provided such as observational (obs.), and known ($\mathcal{K}$) and unknown ($\mathcal{U}$) targets. The last column indicates a brief survey of different fields in ecology, economics, genomics, neurosciences, neurology and medicine that attempt to answer questions at each level. The rows highlighted in "red" are new concepts.

1. **Generalization of standard Markov properties** - We introduce the S-Markov property, which extends and generalizes the normal Markov, $\mathcal{I}$-Markov, and $\Psi$-Markov properties to the setting of multiple domains with arbitrary mixtures of observational and interventional data with known and unknown targets.

2. **Learning algorithm** - We develop a sound learning algorithm for learning an equivalence class of selection diagrams with observational and/or interventional data across different domains.[1]

## 2  Preliminaries and Notation

Uppercase letters ($X$) represent random variables, lowercase letters ($x$) signify assignments, and bold ones ($\mathbf{X}$) indicate sets. The CI relation $\mathbf{X}$ being independent of $\mathbf{Y}$ given $\mathbf{Z}$ is denoted as $\mathbf{X} \perp \mathbf{Y}|\mathbf{Z}$. The d-separation (or m-separation) of $\mathbf{X}$ from $\mathbf{Y}$ given $\mathbf{Z}$ in graph $G$ is expressed as $(\mathbf{X} \perp \mathbf{Y}|\mathbf{Z})_G$. $G_{\overline{X}}$ depicts $G$ with incoming edges to $X$ removed, while $G_{\underline{X}}$ omits all edges outgoing from $X$. Conventionally, every variable is d-separated from the empty set, denoted as $(X \perp \{\})_G$. Superscripts and subscripts will be dropped where feasible to simplify notation.

**Causal Bayesian Network (CBN):** Let $P(\mathbf{V})$ be a probability distribution over a set of variables $\mathbf{V}$, and $P_{\mathbf{x}}(\mathbf{V})$ denote the distribution resulting from the *hard intervention do($\mathbf{X} = \mathbf{x}$)*, which sets $\mathbf{X} \subseteq \mathbf{V}$ to constants $\mathbf{x}$. Let $\mathbf{P}^*$ denote the set of all interventional distributions $P_{\mathbf{x}}(\mathbf{V})$, for all $\mathbf{X} \subseteq \mathbf{V}$, including $P(\mathbf{V})$. A directed acyclic graph (DAG) over $\mathbf{V}$ is said to be a *causal Bayesian network* compatible with $\mathbf{P}^*$ if and only if, for all $\mathbf{X} \subseteq \mathbf{V}$, $P_{\mathbf{x}}(\mathbf{v}) = \prod_{\{i|V_i \notin \mathbf{X}\}} P(v_i|\mathbf{pa}_i)$, for all $\mathbf{v}$ consistent with $\mathbf{x}$, and where $\mathbf{Pa}_i$ is the set of parents of $V_i$ [41, 51, pp. 24]. Given that a subset of the variables are unmeasured or latent, $G(\mathbf{V} \cup \mathbf{L}, \mathbf{E})$ will represent the causal graph where $\mathbf{V}$ and $\mathbf{L}$ denote the measured and latent variables, respectively, and $\mathbf{E}$ denotes the edges. Following the convention in [41], for simplicity, a dashed bi-directed edge is used instead of the corresponding latent variables. CI relations can be read from the graph using a graphical criterion known as *d-separation*.

**Soft Interventions:** Under this type of interventions, the original conditional distributions of the intervened variables $\mathbf{X}$ are replaced with new ones, without completely eliminating the causal effect of the parents. Accordingly, the interventional distribution $P_{\mathbf{X}}(\mathbf{v})$ for $\mathbf{X} \subseteq \mathbf{V}$ is such that $P^*(X_i|Pa_i) \neq P(X_i|Pa_i), \forall X_i \in \mathbf{X}$, and factorizes as follows:

$$P_{\mathbf{X}}(\mathbf{v}) = \sum_{\mathbf{L}} \prod_{\{i|X_i \in \mathbf{X}\}} P^*(x_i|\mathbf{pa}_i) \prod_{\{j|T_j \notin \mathbf{X}\}} P(t_j|\mathbf{pa}_j) \tag{1}$$

In this work, we assume no selection bias and solely consider soft interventions. In the presence of multiple domains, a selection diagram captures commonalities and differences between domains [16, 52, 53]. Represented as $G_S = (\mathbf{V} \cup \mathbf{L} \cup \mathbf{S}, \mathbf{E} \cup \mathbf{E_S})$, it extends a causal diagram by incorporating S-nodes and their edges. $\binom{N}{2}$ S-nodes, $S^{i,j}$, indicate distribution changes across pairs among N domains, by pointing to nodes in $\mathbf{V}$ whose mechanism is altered between domains i and j. An example is shown in Figure 1(a), where the S-node is pointing to $X$, indicating that the distribution of X

---

[1]Our algorithm is implemented in open-source MIT-Licensed https://github.com/py-why/dodiscover.

changes, or that of the latent variable of X is different across the two domains.[2] Similarly, "F-nodes" are auxiliary nodes used in [1, 7, 54] to represent invariances with respect to interventions within the same domain. F-nodes in this paper when written as $F_X^{i,j}$ means it intervenes on X and compares distributions from domains i and j. $F_X^i$ means it compares distributions within domain i. Unlike interventions, domain-shifts potentially alter latent variable distributions or functional relationships and persist irrespective of whether or not external intervention occurs. Distinguishing these concepts enables S-node learning, vital for transportability analysis on ancestral graphs. Appendix Section E.4 elaborates on our distinctions from previous work [11, 13, 14, 36].

Let $\mathbf{S} = \{S^{1,2}, S^{1,3}, ..., S^{N-1,N}\}$ represent $\binom{N}{2}$ S-nodes for distribution changes across domain pairs. When $i = j$, $S^{i,j} = \phi$, indicating there is no S-node for a single domain.

**Multi-domain setup**  The following objects are utilized repeatedly, and introduced here. Our notation borrows from [6] and the transportability literature [55].

1. **Domains**: $\mathbf{\Pi} = \{\Pi^1, \Pi^2, ..., \Pi^N\}$ denotes a set of N domains.
2. **Intervention targets**: $\mathbf{\Psi}^\Pi = \langle \Psi_1^1, \Psi_2^1, ..., \Psi_M^N \rangle$ is an ordered tuple of sets of intervention targets, with different sets of intervention targets occurring within each of the N domains for a total of M intervention target sets. We will denote $\mathbf{\Psi}^i$ as the intervention targets associated with domain i.
3. **Distributions**: $\mathbf{P}^\Pi = \langle P_1^1, P_2^1 ..., P_M^N \rangle$ is an ordered tuple of probability distributions that are available to learn from. Denote $\mathbf{P}^i$ as the distributions associated with domain i. There is a one-to-one correspondence between $\mathbf{P}$ and $\mathbf{\Psi}$, such that $P_j^i$ is the distribution associated with targets $\mathbf{\Psi}_j^i$ in domain i.
4. **Known target indices**: $\mathcal{K}$ is a vector of 1's and 0's indicating which sets of interventions are known-targets. $\mathcal{U} := 1 - \mathcal{K}$ represents therefore an index vector selecting the distributions and interventions with unknown targets. $\mathbf{P}_\mathcal{K}$ and $\mathbf{\Psi}_\mathcal{K}$ denotes the set of distributions and intervention targets corresponding to the known target interventions.
5. **Causal diagram**: $G = (\mathbf{V} \cup \mathbf{L}, \mathbf{E})$, is a shared diagram over the N domains.
6. **Selection diagram**: $G_S = (\mathbf{V} \cup \mathbf{L} \cup \mathbf{S}, \mathbf{E} \cup \mathbf{E_S})$, extends G with the corresponding S-nodes and their edges to represent each pair of domains. Let $\mathbf{V}_{S^{i,j}}$ denote the set of nodes that S-node $S^{i,j}$ points to and $\mathbf{V_S}$ as the set of children for all S-nodes of $G_S$.

$\mathbf{X}^i$ denotes the ith domain set of variables $\mathbf{X}$, and $X_i \in \mathbf{X}$ indicates the ith variable within $\mathbf{X}$. When discussing intervention targets, $X_j^{i,(k)}$ refers to the jth variable with the kth mechanism change in domain i. For instance, $X^{i,(k)}, X^{i,(l)}$ represent two interventions with distinct mechanisms (k and l) on variable X in domain i. $\{\}^i \in \mathbf{\Psi}$ explicitly denotes the observational distribution for domain i and is by convention a "known-target". For concreteness, say $\mathbf{\Pi} = \{\Pi^1, \Pi^2, \Pi^3\}$ with $\mathbf{P} = \langle P_1^1, P_2^1, P_3^1, P_1^3 \rangle$, $\mathbf{\Psi} = \langle \{\}^1, \{X^{(a)}\}^1, \{X, Y\}^1, \{\}^3 \rangle$, and $\mathcal{K} = [1, 1, 0, 1]$. In words, there are three distributions available in domain 1: $P_1^1$ is observational, $P_2^1$ is known-target on X with a specific mechanism change and $P_3^1$ is unknown-target that intervenes on X and Y simultaneously. In domain 3, $P_1^3$ is observational. There are no distributions for domain 2.

## 3   Multi-domain Markov Equivalence Class

Before designing a learning algorithm, we must characterize what can be learned from the given selection diagram. This section explores ECs in a multi-domain setting with arbitrary mixtures of observational and interventional data. The following assumptions are made throughout the paper.

**Assumption 1** (Shared causal structure). We assume that each environment shares the same causal diagram. That is the S-nodes do not change the underlying causal diagram.  □

This means that the S-nodes do not represent structural changes such as when $V_i$ has a different parent set across domains.[3]

---

[2]In the original selection diagram, each S-node points to a single node. Our adaptation simplifies it to a single S-node with multiple connections. Theoretical properties remain unaffected, as shown in the appendix.

[3]The assumption that there are no structural changes between domains can be relaxed in the context of inference, as specified in [16]. We do not explore this relaxation here in the context of structure learning.

**Assumption 2** (Observational data is present across domains). We make the simplifying assumption that $\{\} \in \Psi^i$, $\forall i \in [N]$, that is observational data is present in all domains.

This is a realistic assumption in many scientific applications highlighted in Table 1.[4] Another assumption we make is that all soft interventions across domains are *distinct*.

**Assumption 3** (Distinct interventions across domains). We assume all the interventions across different domains have unique mechanisms. That is, if $X^{(m)} \in \Psi^i$ and $X^{(n)} \in \Psi^j$, where $i \neq j$, then $m \neq n$. In words, $X$ has different mechanisms across the two distributions $P_{\Psi^i}, P_{\Psi^j}$.

This is a realistic assumption that precludes the possibility that any interventions that occur in different domains result in the same exact mechanism. For example, even if medication is given to humans and bonobos, it is unrealistic to expect the intervention has the same mechanism of action in each domain. Next, we define an important operation when comparing two different intervention sets.

**Definition 3.1** (Symmetrical Difference Operator $\Delta$ in Multiple Domains). For two domains $i, j$ (possibly $i = j$), given two sets of intervention targets, $\Psi^i$ and $\Psi^j$, let $\Psi^i \Delta \Psi^j$ denote the symmetrical difference set such that $X \in \Psi^i \Delta \Psi^j$ if $X^{(k)} \in \Psi^i$ and $X^{(k)} \notin \Psi^j$ or vice versa. $\qquad \square$

This operation identifies the set of variables with unique interventional mechanisms across two intervention targets and also tracks the domain ids. For example, given $\mathbf{I}^1 = \{X^1, Y, Z\}^1$ and $\mathbf{J}^1 = \{X^2, Y\}^2$, then $\mathbf{I}^1 \Delta \mathbf{J}^1 = \{X, Z\}^{1,2}$. An implication of the above definition and Assumption 3 is that the symmetrical difference of two intervention target sets from two different domains is the union of all the variables in both sets since the mechanisms would be unique. For more details and discussion on the assumptions, see the Appendix.

## 3.1 Multi-distributional invariances: interventions and change-of-domain

This section elaborates on exactly what type of distributional invariances we characterize in the so called S-Markov EC.

When given only observational data, the celebrated FCI algorithm uses invariances of the form $P(\mathbf{Y}|\mathbf{X}, \mathbf{W}) = P(\mathbf{Y}|\mathbf{X})$ within the same probability distribution $P(\mathbf{V})$ to characterize the Markov EC [2]. These invariances, or CI statements can be mapped to d-separation statements in the graphical model. The resulting learned object is the PAG, which represents the EC when only observational data is given within a single domain.

The works in [6–8, 44] build upon the Markov EC to characterize the so called interventional Markov EC, which uses distributional invariances of the form $P_{\mathbf{W}}(\mathbf{Y}|\mathbf{X}) = P_{\mathbf{Z}}(\mathbf{Y}|\mathbf{X})$. In words, these are conditional probabilities that remain invariant under different interventions. Importantly, this sort of invariance is markedly different from that of the CI statements, where only observational data is present, because one is now comparing probabilities across *different distributions*. These distributional invariances can be characterized graphically by the d-separation property when using an augmented graph with "F-nodes", which serve as graphical representations of the differences in distributions due to interventions. However, this body of work assumes that all the distributions, observational and interventional, are within the *same* domain.

In this work, we generalize this setting and consider an input set of distributions from (possibly) different domains to characterize the S-Markov EC. We consider distributional invariances of the form $P_{\mathbf{W}}^i(\mathbf{Y}|\mathbf{X}) = P_{\mathbf{K}}^j(\mathbf{Y}|\mathbf{X})$ such that distributions could stem from different domains when $i \neq j$. Such an invariance implies that the conditional distribution of $\mathbf{Y}|\mathbf{X}$ remains the same across domains $i$ and $j$ under interventions on $\mathbf{W}$ and $\mathbf{K}$, respectively. Whenever $i = j$, these invariances reduce to the ones considered in the interventional Markov EC. From this perspective, it is clear that multi-domain invariances generalize the invariances analyzed in observational and interventional data in a single-domain.

## 3.2 S-Markov Property

Now, we are ready to generalize the previous Markov properties [2, 5–7, 41, 56] to the case when observational, and known/unknown-target interventional distributions in multiple domains are available.

---

[4]If one can collect experimental data in a domain, it is reasonable that they can also collect observational data. We discuss this further in the Appendix.

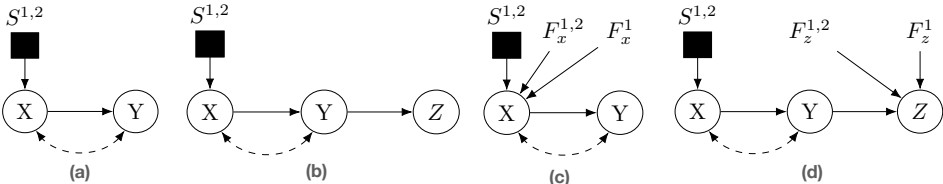

Figure 1: Example selection diagrams (a,b) and their respective augmented graphs (c,d). (c) corresponds to intervention set $\mathbf{\Psi} = \langle\{\}^1, \{\}^2, \{X\}^1\rangle$ and (d) corresponds to intervention set $\mathbf{\Psi} = \langle\{\}^1, \{\}^2, \{Z\}^1\rangle$.

**Definition 3.2** (S-Markov Property). Given the Multi-domain setup, $\mathbf{P}$ satisfies the S-Markov property with respect to the pair $\langle G_S, \mathbf{\Psi}\rangle$ if the following holds for disjoint $\mathbf{Y}, \mathbf{W}, \mathbf{Z} \subseteq \mathbf{V}$:

1. For $\mathbf{\Psi}_j^i \in \mathbf{\Psi}$: $\qquad P_j^i(\mathbf{y}|\mathbf{w}, \mathbf{z}) = P_j^i(\mathbf{y}|\mathbf{w}) \quad$ if $(\mathbf{Y} \perp \mathbf{Z}|\mathbf{W}, \mathbf{S})_{G_S}$

2. For $\mathbf{\Psi}_m^i, \mathbf{\Psi}_l^j \in \mathbf{\Psi}$: $\quad P_m^i(\mathbf{y}|\mathbf{w}) = P_l^j(\mathbf{y}|\mathbf{w}) \quad$ if $(\mathbf{Y} \perp \mathbf{K}|\mathbf{W}\backslash\mathbf{W_K}, \mathbf{S} \setminus \{S^{i,j}\})_{G_{S_{\underline{\mathbf{W_K}}, \overline{\mathbf{R(W)}}}}}$,

where $\mathbf{K} = (\mathbf{\Psi}_m^i \Delta \mathbf{\Psi}_l^j) \cup \{S^{i,j}\}$, $\mathbf{W_K} = \mathbf{W} \cap \mathbf{K}$, $\mathbf{R} = \mathbf{K}\backslash\mathbf{W_K}$ and $\mathbf{R(W)} \subseteq \mathbf{R}$ are non-ancestors of $\mathbf{W}$ in $G_S$.

Let $S_{\mathcal{K}}^{\Pi}(G_S, \mathbf{\Psi})$ denote the set of distribution tuples that satisfy the S-Markov property with respect to $\langle G_S, \mathbf{\Psi}\rangle$ where $\mathcal{K}$ denotes the known intervention targets. $\qquad\square$

When there is only a single domain, $\mathbf{\Pi} = \{\Pi^1\}$, the first constraint reduces to standard d-separation on a causal diagram. The second condition is a generalization of the $\Psi$-Markov property characterization [6], extending conditional invariances to multiple domains. We illustrate the definition with the following two examples.

**Example 1.** Consider the selection diagram in Figure 1(a) with two domains $\mathbf{\Pi} = \{\Pi^1, \Pi^2\}$. Let $\mathbf{P} = \langle P_1^1, P_2^1, P_1^2\rangle$ be the result of the interventions $\mathbf{\Psi}^{\Pi} = \langle\{\}^1, \{X\}^1, \{\}^2\rangle$, $\mathbf{S} = \{S^{1,2}\}$ be the set of S-nodes, and $\mathcal{K} = [1, 0, 1]$. First, we have $(Y \not\perp X|S^{1,2})_{G_S}$ so the first constraint of Def. 3.2 is not applicable for any distribution. Second, we compare $P_1^1(y|x)$ and $P_2^1(y|x)$, where $S^{1,1} = \emptyset$ by convention and $\mathbf{K} = \{X\}$. We have $(Y \not\perp X)_{G_{S_{\underline{X}}}}$ and the invariance is not required. For $P_1^1(y|x)$ and $P_1^2(y|x)$, we have $(Y \not\perp S|X)_{G_S}$. Also, for $P_2^1(y|x)$ and $P_1^2(y|x)$, we have $(Y \not\perp \{X, S\})_{G_{S_{\underline{X}}}}$. Hence, no invariances are required between those pairs of distributions. A similar argument can be made when comparing other probability terms across distributions. Therefore, $\mathbf{P}$ satisfies the S-Markov property with respect to $\langle G_S, \mathbf{\Psi}\rangle$. $\qquad\square$

**Example 2.** Consider the setup from Ex. 1. We check if $\mathbf{P}$ satisfies the S-Markov property relative to $\langle G_S, \mathbf{\Psi}'\rangle$ where $\mathbf{\Psi}'^{\Pi} = \langle\{\}^1, \{Y\}^1\{\}^2\rangle$. We compare $P_1^1(X)$ and $P_2^1(X)$ and we have $K = (\{\}^1 \Delta \{Y\}^1) \cup \emptyset = \{Y\}^1$. The separation $(X \perp Y|S^{1,2})_{G_{\overline{Y}}}$ holds true which implies the invariance $P_1^1(X) = P_2^1(X)$, but $P_2^1$ was generated from an intervention on $X$ so the invariance is not satisfied. Hence, $\mathbf{P}$ does not satisfy the S-Markov property with respect to $\langle G_S, \mathbf{\Psi}'\rangle$. $\qquad\square$

Next, we use Def. 3.2 to define S-Markov equivalence as follows. In words, two pairs of selection diagrams and their corresponding sets of intervention targets $\langle G_S, \mathbf{\Psi}\rangle$ and $\langle G_S', \mathbf{\Psi}'\rangle$ are S-Markov equivalent if they can induce the same set of distribution tuples.

**Definition 3.3** (S-Markov Equivalence). Let $\mathbf{\Pi}$ and $\mathcal{K}$ denote fixed sets of domains and indices of known intervention targets, respectively. Given selection diagrams $G_S, G_S'$ defined over $\mathbf{V} \cup \mathbf{S}$ and the corresponding intervention targets $\mathbf{\Psi}, \mathbf{\Psi}'$, the pairs $\langle G_S, \mathbf{\Psi}\rangle$ and $\langle G_S', \mathbf{\Psi}'\rangle$ are said to be S-Markov equivalent if $S_{\mathcal{K}}^{\Pi}(G_S, \mathbf{\Psi}) = S_{\mathcal{K}}^{\Pi}(G_S', \mathbf{\Psi}')$. $\qquad\square$

### 3.3 Multi-domain observational data

S-nodes introduced through the lens of selection diagrams are augmentations of the causal graph to represent different domains and changes in distributions that may occur [7, 15, 54, 57]. As part of this augmented graph, S-nodes are graphically similar to F-nodes, which have been successfully used to represent interventions [6, 7, 54]. F-nodes are utility nodes where each one is a parent to (each element

in) a symmetrical difference set, and they are used to represent invariances between interventional distributions. The significance of these F-nodes will be further emphasized in Section 3.4; more specifically, by Definition 3.5 and Proposition 1. Despite the similarity between F-nodes and S-nodes, it is worthy to distinguish S-nodes since many causal inference tasks, such as in transportability, rely on knowing the S-node structure [15, 53]. Before deriving the graphical characterization for the S-Markov equivalence class, we first focus on the setting where there is only observational data across different domains. We highlight that S-nodes can be exactly viewed as F-nodes constructed from interventions with unknown targets when there is only observational data to consider [6].

**Definition 3.4** (Corresponding Intervention Set). Consider the Multi-domain setup. For a selection diagram $G_S$ over N domains. $\langle \mathbf{V}_{S^{i,j}} \rangle = \langle \mathbf{V}_{S^{1,2}}, \mathbf{V}_{S^{1,3}}, ..., \mathbf{V}_{S^{N-1,N}}, \rangle \forall i \neq j \in [N]$ is an ordered tuple of the children of each S-node. The corresponding intervention set for $\mathbf{V_S}$ is $\langle \mathbf{I}^1, \mathbf{I}^2, ..., \mathbf{I}^N \rangle$, such that $\mathbf{I}^i \Delta \mathbf{I}^j = \mathbf{V}_{S^{i,j}}$ for all $i \neq j$. □

The corresponding intervention set is a set that simplifies our presentation of the following theorem.

**Theorem 1** (Equivalence of $\Psi$ and S Markov property given multi-domain observational distributions). Consider the Multi-domain setup. Let $G_S$ be a selection diagram among N domains and G be the corresponding causal diagram without S-nodes. Let $\mathbf{\Psi}^\Pi = \langle \{\}^1, ..., \{\}^N \rangle$ and $\mathcal{K} = [1, 1, ..., 1]$, such that for each of the N domains, there is only observational data. Let $\mathbf{I}_S$ be the corresponding intervention set for $\mathbf{V_S}$. Let $\mathbf{P}^\Pi$ be an arbitrary set of distributions generated by the corresponding interventions. $\mathbf{P}^\Pi$ satisfies the S-Markov property with respect to $\langle G_S, \mathbf{\Psi} \rangle$ if and only if it satisfies the $\Psi$-Markov property with respect to $\langle G, \mathbf{I}_S \rangle$.[5] □

When given observations collected from multiple domains, it is equivalent to collecting distributions with unknown-target interventions. This coincides with other works, which treat different domains and interventions as the same [10, 13]. In this setting, S-nodes have a correspondence to the augmented graph's F-nodes in [6]. In some sense, the change-in-domain can be viewed as "nature's" intervention on the causal system. However, this simplification is not warranted when we consider interventions that occur in different domains.

### 3.4 Mixture of multi-domain observational and interventional data

Next, we analyze the general setting with multi-domain observational and interventional data. Def. 3.2 and 3.3 may be quite challenging to evaluate in practice since it involves surgically altering the selection diagram. One can leverage a graphical approach that encodes the symmetric differences of interventions using F-nodes [7].

**Definition 3.5** (Augmented selection diagram). Consider the Multi-domain setup. Let the multiset $\mathcal{I}$ be defined as $\mathcal{I} = \{\mathbf{K}_1, \mathbf{K}_2, ...\mathbf{K}_k\} = \{\mathbf{K} | \Psi_m^i, \Psi_l^j \in \mathbf{\Psi} \wedge \Psi_m^i \Delta \Psi_l^j = \mathbf{K}\}$. The augmented graph of $G_S$ with respect to $\mathbf{\Psi}$ is denoted as $Aug_{\mathbf{\Psi}}(G_S)$ and constructed as follows: $Aug_{\mathbf{\Psi}}(G_S) = (\mathbf{V} \cup \mathbf{L} \cup \mathbf{S} \cup \mathcal{F}, \mathbf{E} \cup \mathbf{E_S} \cup \mathcal{E})$, where $\mathcal{F} = \{F_i^{j,k}\}^{j,k \in [N]}$ is the set of added F-nodes and $\mathcal{E} = \{(F_i^{j,k}, l)\}_{l \in \mathbf{K}_i}$ is the set of added F-node edges. □

The F-nodes graphically encode the symmetrical difference sets between every pair of intervention targets in $\mathbf{\Psi}^\Pi$ (i.e. $\Psi_m^i \Delta \Psi_l^j$) within and across the different domains in $\Pi$. $F_k^{i,i} = F_k^i$ denotes an F-node representing the kth symmetric difference of intervention targets within domain $i$ and $F_k^{i,j}$ denotes an F-node from comparing intervention targets between domains $i$ and $j$. The result is an augmented selection diagram with the original causal structure augmented with S-nodes, F-nodes, and their additional edges. For example, Figure 1(c) shows the augmented diagram of the selection diagram in Figure 1(a) with respect to $\mathbf{\Psi}^\Pi = \langle \{\}^1, \{X\}^1, \{\}^2 \rangle$. The significance of this construction follows from Proposition 1 where separation statements in the S-Markov definition are tied (shown to be equivalent, formally speaking) to ones in the augmented selection diagram, with no need to perform any graphical manipulations.

**Proposition 1** (Graphical S-Markov Property). Consider the Multi-domain setup. Let $Aug_{\mathbf{\Psi}}(G_S)$ be the augmented graph of $G_S$ with respect to $\mathbf{\Psi}$. Let $\mathbf{K}_i^{j,k} = \mathbf{K}_i \cup \{S^{j,k}\}$ be the union of the set of nodes adjacent to $F_i^{j,k}$ and the corresponding S-node $S^{j,k}$. The following equivalence relations hold

---

[5]Due to space constraints, all the proofs are provided in the Appendix.

for disjoint $\mathbf{Y}, \mathbf{Z}, \mathbf{W} \subseteq \mathbf{V}$, where $\mathbf{W}_i = \mathbf{W} \cap \mathbf{K}_i$ and $\mathbf{R} = \mathbf{K}_i \backslash \mathbf{W}_i$.

$$(\mathbf{Y} \perp \mathbf{Z} | \mathbf{W}, \mathbf{S})_{G_S} \iff (\mathbf{Y} \perp \mathbf{Z} | \mathbf{W}, \mathbf{S}, \mathcal{F})_{Aug_{\mathbf{\Psi}}(G_S)} \tag{2}$$

$$(\mathbf{Y} \perp \mathbf{K}_i^{j,k} | \mathbf{W} \backslash \mathbf{W}_i, \mathbf{S} \backslash \{S^{j,k}\})_{G_{S_{\underline{\mathbf{W}_i}, \overline{\mathbf{R}(\mathbf{W})}}}} \iff (\mathbf{Y} \perp \{F_i^{j,k}, S^{j,k}\} | \mathbf{W}, F_{[k] \backslash \{i\}}, \mathbf{S} \backslash \{S^{j,k}\})_{Aug_{\mathbf{\Psi}}(G_S)} \tag{3}$$

$\square$

The result in the above proposition is illustrated in the following example.

**Example 3.** Consider the selection diagram in Fig. 1(b) with intervention targets $\mathbf{\Psi} = \langle \emptyset^1, \{Z\}^1, \emptyset^2 \rangle$. By Prop. 1, we can evaluate the S-Markov property in the corresponding augmented diagram in Fig. 1(d) without manipulating it. For example, $(Y \perp Z)_{G_{S_{\overline{Z}}}}$ can be tested in Fig. 1(d) by $(Y \perp F_z^1 | \mathbf{S}, F_z^{1,2})_{Aug_{\mathbf{\Psi}}(G_S)}$ to determine if the invariance $P^1(Y) = P_Z^1(Y)$ should hold. In addition, we can test if across-domain distributional invariances should hold. We have $(Y \not\perp \{F_z^{1,2}, S^{1,2}\} | \{X, Z\}, F_z^1)_{Aug_{\mathbf{\Psi}}(G_S)}$, then the invariance $P_Z^1(Y|X, Z) = P^2(Y|X, Z)$ is not required. $\square$

Maximal Ancestral Graphs (MAGs) provide a compact and convenient representation capable of preserving all the tested constraints in augmented selection diagrams which are represented by d-separations [58]; see also [59, p. 6]. This is formalized in Definition 3.6 and the construct is referred to as an *S-MAG*. The following example is provided for illustration.

**Definition 3.6** (S-MAG). Given a selection diagram $G_S$ and a set of intervention targets $\mathbf{\Psi}$, an S-MAG is the MAG constructed from $Aug_{\mathbf{\Psi}}(G_S)$. That is $\text{MAG}(Aug_{\mathbf{\Psi}}(G_S))$. $\square$

**Example 4.** Consider the selection diagram in Figure 1(a) and let $\mathbf{\Psi} = \langle \{\}^1, \{X\}^1, \{\}^2 \rangle$. The corresponding augmented selection diagram $Aug_{\mathbf{\Psi}}(G_S)$ is shown in Fig. 1(c). Finally, the corresponding S-MAG is $MAG(Aug_{\mathbf{\Psi}}(G_S)) = \{X \leftarrow F_x^1 \rightarrow Y, X \leftarrow F_x^{1,2} \rightarrow Y, X \leftarrow S^{1,2} \rightarrow Y, X \rightarrow Y\}$. $\square$

Finally, putting these results together, we derive a graphical characterization for two selection diagrams with corresponding sets of intervention targets to be S-Markov equivalent.

**Theorem 2** (S-Markov Characterization). Let $\mathbf{\Pi}$ and $\mathcal{K}$ denote fixed sets of domains and indices of known intervention targets, respectively. Given selection diagrams $G_S, G_S'$ defined over $\mathbf{V} \cup \mathbf{S}$ and the corresponding intervention targets $\mathbf{\Psi}, \mathbf{\Psi}'$, the pairs $\langle G_S, \mathbf{\Psi} \rangle$ and $\langle G_S', \mathbf{\Psi}' \rangle$ are S-Markov equivalent if and only if for $M = MAG(Aug_{\mathbf{\Psi}}(G_S))$ and $M' = MAG(Aug_{\mathbf{\Psi}'}(G_S'))$:[6]

1. $M$ and $M'$ have the same skeleton;

2. $M$ and $M'$ have the same unshielded colliders; and,

3. If a path $p$ is a discriminating path for a node $Y$ in both $M$ and $M'$, then $Y$ is a collider on the path in one graph if and only if it is a collider on the path in the other. $\square$

Theorem 2 states that the pairs $\langle G_S, \mathbf{\Psi} \rangle$ and $\langle G_S', \mathbf{\Psi}' \rangle$ are S-Markov equivalent if their corresponding S-MAGs satisfy the corresponding three conditions, as illustrated in the example below.

**Example 5.** Consider the tuples $\langle G_S, \mathbf{\Psi}^{\mathbf{\Pi}} \rangle$ from Example 1 and $\langle G_S, \mathbf{\Psi}'^{\mathbf{\Pi}} \rangle$ from Example 2. S-MAGs $M = MAG(Aug_{\mathbf{\Psi}}(G_S))$ is shown in Ex. 4 and $M' = MAG(Aug_{\mathbf{\Psi}'}(G_S)) = \{F_y^1 \rightarrow Y, F_y^{1,2} \rightarrow Y, X \leftarrow S^{1,2} \rightarrow Y, X \rightarrow Y\}$. Therefore, $M_1$ and $M_2$ have differing skeletons and thus are not S-Markov equivalent. $\square$

Next, we leverage this characterization to devise an algorithm to learn the corresponding equivalence class of a true, underlying selection diagram.

## 4 Causal Discovery From Multiple Domains

We investigate in this section how to learn an EC of selection diagrams from a mixture of observational and interventional data that is generated from multiple domains. The graphical characterization of S-Markov equivalence in Theorem 2 and the significance of ancestral graphs (MAGs) in deriving this result motivate the following definition of S-PAG.

---

[6]We assume that the symmetrical difference sets are indexed for both diagrams in the same pattern such that the correspondence between F-nodes and S-nodes is the same in $M$ and $M'$.

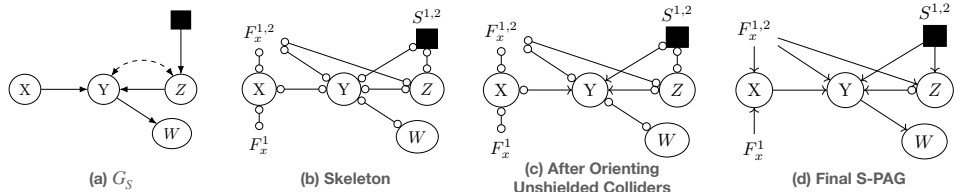

Figure 2: Example of S-FCI applied with $\boldsymbol{\Psi} = \langle \{\}^1, \{X\}^1, \{\}^2 \rangle$ and $\mathcal{K} = [1, 1, 1]$. The S-node representing domain-shift between domains 1 and 2 is the black square in (a).

---

**Algorithm 1 S-FCI: Algorithm for Learning a S-PAG** - $SepSet$ the separating sets, $\mathbf{S}$ is the S-node set, $\mathcal{F}^\Pi$ the F-node set, $\mathbf{H}$ maps each pair of known-targets symmetric diffs., and $\sigma$ maps each pair of distributions to a pair of domains.

---

**Input:** Tuple of distributions $\mathbf{P}^\Pi = \langle P_1^1, ..., P_m^N \rangle$, vector of known intervention targets $\mathcal{K}$ and $\boldsymbol{\Psi}^\Pi$.
**Output:** S-PAG, $\mathcal{P}$
   $\mathbf{S}, \mathcal{F} \leftarrow \phi, k \leftarrow 0, \sigma : \mathbb{N} \rightarrow \mathbb{N} \times \mathbb{N}, \mathbf{H} \leftarrow \phi$
   $(\mathbf{S}, \mathcal{F}, \mathbf{H}, \sigma) \leftarrow \text{CreateAugmentedNodes}(\boldsymbol{\Psi}^\Pi, V)$ (see Alg. E.2)
   **Phase I: Learn skeleton**
   **for** all pairs $X, Y \in \mathbf{V} \cup \mathcal{F} \cup \mathbf{S}$ **do**
      $SepSet(X, Y), SepFlag \leftarrow \text{GeneralizedDoConstraints}(X, Y, \mathcal{F}, \mathbf{S}, \sigma, \boldsymbol{\Psi}^\Pi, \mathcal{K}, \mathbf{V})$ (see Alg. E.4)
      **if** SepFlag = True **then**
         Remove edge between X and Y
   **Phase IIa: Orient unshielded colliders**
   For every unshielded triple $\langle X, Y, Z \rangle$ in $\mathcal{P}$ orient it as a collider iff $Z \notin SepSet(X, Y)$
   **Phase IIb: Apply logical orientation rules**
   R1-7: Apply 7 FCI rules from [39] and following two rules until none apply.
   Rule 8': For $F_k^{i,j} \in \mathcal{F}^\Pi$ and for $S^{i,j} \in \mathbf{S}$, orient adjacent edges out of $F_k^{i,j}$ and $S^{i,j}$.
   Rule 9': For $F_k^{i,j} \in \mathcal{F}^\Pi$ with $X \in H_k^{i,j}$, that is adjacent to a node $Y \notin H_k^{i,j}$, if $|H_k^{i,j}| = 1$, then orient $X \rightarrow Y$.

---

**Definition 4.1** (S-PAG). Consider the Multi-domain setup. Let $M = MAG(Aug_{\boldsymbol{\Psi}}(G_S))$ and let $[M]$ be the set of S-MAGs corresponding to all the tuples $\langle G'_S, \boldsymbol{\Psi}'^\Pi \rangle$ that are S-Markov equivalent to $\langle G_S, \boldsymbol{\Psi}^\Pi \rangle$. The S-PAG for $\langle G_S, \boldsymbol{\Psi}^\Pi \rangle$, denoted $\mathcal{P}$ is a graph such that:

1. $\mathcal{P}$ has the same adjacencies as M and any member of [M] does; and

2. every non-circle mark (tail or arrowhead) in $\mathcal{P}$ is an invariant mark in [M] (i.e. present in all the S-MAGs in [M]). □

S-PAGs generalize PAGs and $\Psi$-PAGs from the single-domain to the multiple-domain setting[6, 42]. The S-nodes and F-nodes are not so much "random variables" as they are graphical instruments that encode differences across domains and among interventional distributions in this equivalence class. Next, we introduce a generalization of c-faithfulness [6] that enables causal discovery from multi-domain data.

**Definition 4.2** (S-faithfulness). Consider a selection diagram $G_S$ over N domains. A tuple of distributions $\langle \mathbf{P_I} \rangle_{\mathbf{I} \in \boldsymbol{\Psi}^\Pi} \in S_\mathcal{K}^\Pi(G_S, \boldsymbol{\Psi}^\Pi)$ is called s-faithful to $G_S$ if the converse of each of the S-Markov conditions (Definition 3.2) holds. □

The new algorithm, called S-FCI is shown in Alg. 1. Due to space constraints, we only include the high-level algorithm here. The algorithm proceeds by first constructing the augmented graph using Alg. E.2, by adding S-nodes and F-nodes to represent every pair of domains and interventions. Then it uses hypothesis testing to learn invariances in the skeleton (Alg. E.3) and finally applies orientation rules (Alg. E.5). S-FCI learns the skeleton by mapping pairs of distributions in $\mathbf{P}^\Pi$ to F-nodes, or S-nodes by testing for the distributional invariances discussed in Section 3.1. Def. 3.2 and Prop. 1 connect these invariances to graphical criterion, which allow us to reconstruct the skeleton of the causal diagram. Interventional distributions across domains are used to learn F-node structure, and

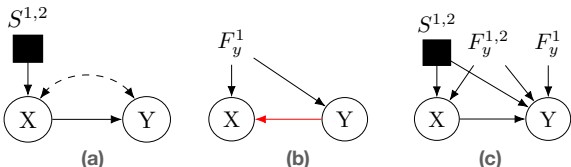

Figure 3: Causal graphs related to example 6 - selection diagram with an intervention at Y, and S-node pointing to X (a), the graph after applying $\mathcal{I}$-FCI without considering domain-changes (b) and the S-PAG learned by S-FCI (c).

whereas observational distributions across domains are used to learn S-node structure. Besides the standard FCI rules that apply in the absence of selection bias, the algorithm also applies the following rules R8'-9'.

**Rule 8' (Augmented Node Edges)** - We orient edges out of F-nodes.

**Rule 9' (Identifiable Inducing Paths)** - If $F_k^{i,j} \in \mathcal{F}$ is adjacent to a $Y \notin H_k^{i,j}$ known-target node and we know that the intervention target is node X, one can orient $X \to Y$ because the $F_k^{i,j} \to Y$ is only present due to an inducing path between X and Y.

In Figure 2, the different stages of the S-FCI algorithm are shown. Next we prove the proposed S-FCI algorithm is sound.

**Theorem 3** (S-FCI Soundness). Given $\mathcal{K}$, let $\mathbf{P}^{\Pi}$ be generated by some unknown tuple $\langle G_S, \mathbf{\Psi}^{\Pi} \rangle$ from domains $\mathbf{\Pi}$ with a corresponding selection diagram $G_S$ and is s-faithful to the selection diagram $G_S$. S-FCI algorithm is sound (i.e. every adjacency and orientation in $\mathcal{P}_{\text{S-FCI}}$, the S-PAG learned by S-FCI, is common for $MAG(Aug_{\Psi}(G_S))$). $\square$

Next, we illustrate some subtleties between the S-FCI and related algorithms that say pool observational and interventional distributions, ignoring the domain change. The example is motivated from biomedical sciences, where interventions are commonly performed in different domains and the goal is to leverage all datasets for learning. A group of scientists are trying to determine the causal structure of a set of proteins, but leverage data across the lab and hospital setting. Different experiments are run in each setting and combined into a single dataset [29]. We provide additional examples and commentary on the S-FCI subtleties in the Appendix.

**Example 6.** Let $G_S$ be a selection diagram as shown in Figure 3(a). Let $\mathbf{\Pi} = \langle \Pi^1, \Pi^2 \rangle$ be the set of domains representing the lab ($\Pi^1$) and the hospital ($\Pi^2$). These are a tuple of distributions $\mathbf{P} = \langle P_1^1, P_1^2 \rangle$ with intervention targets $\mathbf{\Psi}^{\Pi} = \langle \{\}^1, \{Y\}^1, \{\}^2 \rangle$ and $\mathcal{K} = [1, 1, 1]$, where X represents some protein in the dataset.

In this example, let $G_S$ be the true selection diagram as shown in Figure 3(a). Given the interventional and observational data, we may be tempted to use the $\mathcal{I}$-FCI algorithm and simply pool the observational data, while ignoring the domain differences [7], but this would learn the graph in Figure 3(b) with an incorrect orientation (shown as the red edge). This I-PAG only contains one F-node because there is only two distributions: i) the pooled observational data and ii) the data resulting from intervention on Y. Applying R9 of the $\mathcal{I}$-FCI algorithm incorrectly orients the edge $X \leftarrow Y$. Thus, R9 of the $\mathcal{I}$-FCI algorithm is not sound when the domains are ignored [7, 44].

Figure 3(c) contains what S-FCI would recover. Intuitively, one should learn (c) instead of (b) because even though there is a change in distribution among X and Y, one cannot ascertain whether there is an inducing path from $F_y^1$ to X, or a change in distribution due to the domain. $\square$

# 5 Conclusions

In this paper, we introduced a generalized Markov property called S-Markov, which defines a new equivalence class (EC), the S-PAG, representing the constraints found across observational and experimental distributions collected from multiple domains. Building on this new characterization, we develop a causal discovery algorithm called S-FCI, which subsumes FCI, $\mathcal{I}$-FCI and $\Psi$-FCI, and accepts as input a mixture of observational and interventional data from multiple domains.

## Acknowledgements

AL was supported by the NSF Computing Innovation Fellowship (#2127309). EB was supported in part by the NSF, ONR, AFOSR, DoE, Amazon, JP Morgan, and The Alfred P. Sloan Foundation.

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
