# OpenReview forum: "Causal discovery from observational and interventional data across multiple environments"
_NeurIPS.cc/2023/Conference — NeurIPS 2023 poster_

### Official Review · Reviewer_9ikg · 2023-06-29

**Soundness:** 3 good
**Presentation:** 2 fair
**Contribution:** 2 fair
**Rating:** 5
**Confidence:** 2

**Summary:**

This paper discusses the multi-domain scenario for causal discovery, instead of the classic single-domain scenario. It figures out the object to learn for this the muti-domain causal discovery task and proposes corresponding methods for optimizing S-PAGs. It extends existing Markov properties to the S-Markov property, which is more compatible to the multi-domain setting. An algorithm for learning MEC under this diagram is proposed, and the author claims it will be implemented as an open-source repo.

**Strengths:**

i. This paper discusses a new setting, i.e., a mixture of observational and interventional data from multiple domains. This setting is meaningful, which can be used in many real-world applications.
ii. This paper proposes the definition of the multi-domain Markov equivalence class and provides the method of causal discovery from multiple domains.
iii. Many examples are provided in Section 2 to help with explanations.

**Weaknesses:**

i. The symbols are a bit disordered, which makes readers hard to understand and follow. A clearer definition in Section 1 is possibly needed.
ii. No relation works, which may increase the hardness for readers.
iii. The experiments are supposed to be placed in the main content, which is important for this paper. The structure of this paper may be re-considered.
iv. Many theorems and rules are proposed, but it seems that the proof is not mentioned.

**Questions:**

i. What about the efficiency of the proposed algorithm?
ii. Are the assumptions on the multi-domain setting too strong? How could we relax them?

**Limitations:**

See weakness

---

> ### Author Rebuttal · Authors · 2023-08-10
>
> > weakness-“i. The symbols are a bit disordered, which makes readers hard to understand and follow.”
>
> We’ve experimented with various notations but they tend to be somewhat heavy and we apologize. This complexity arises because the novel characterization and learning algorithm necessitate a formal distinction between distributions, domains, and interventions, which requires keeping track of distributions, intervention sets, domain IDs, known-targets, etc.
>
> In light of your feedback, we’ve restructured the preliminary section and introduced symbols in a bullet-point format for easier comprehension of these concepts (shown in the general author rebuttal). We also dropped the subscript (set index) and superscript (domain index) in the rest of the paper where possible to make the reading simpler.
>
> >  w-“ii. No relation works, which may increase the hardness for readers.”
>
> We respectfully disagree. In the original submission, we referenced various works in the introduction that present characterizations for ancestral graphs, including the I-Markov equivalence class (EC) and the $\Psi$-Markov EC (lines 50-54). Specifically, we highlighted works that characterize the Markov EC in contexts like observational (obs) data alone and combined observational and interventional data. We also touched upon other works in the context of “multiple domains” (L55-61).
>
> A deeper comparison with other works is provided in Appendix Section “D.2 Comparisons with Other Works”. Additionally, Table 1 cites studies in the sciences that seek to answer causal discovery questions in the settings discussed. Having said that, if there are important references you believe we might have overlooked, we would be glad to include them.
>
> > w-“iii. The experiments are supposed to be placed in the main content, which is important for this paper.”
>
> Given the pg. limit and the introduction of theoretical concepts in the main text, we opted to include relevant experiments in the Appendix, specifically in “Section D.3 Experimental Results - Simulations.” In the original submission, we presented an experiment in this section (L1133-1168) that demonstrated the incorrect orientation of edges when one fails to account for interventions occurring in different domains. This is illustrated with the result of S-FCI (Figure S5 in the original submission) on multi-distributional data, such as the Sachs dataset.
>
> Furthermore, we introduce a simulation in Figure S10 (see attached fig. PDF) comparing the graphs learned by FCI, $\Psi$-FCI, and S-FCI algorithms. By recognizing data from different domains, the S-FCI algorithm can orient more true edges in alignment with the true graph than other FCI and $\Psi$-FCI strategies using the same dataset.
>
> We understand that the placement of experiments in a paper may vary according to authors' preferences, so we respectfully disagree with the notion that there's a universal standard for their placement in the main body. Having said that, we're very open to rearranging content if the reviewers agree that this is the most suitable option.
>
> > w-”iv. Many theorems and rules are proposed, but it seems that the proof is not mentioned.”
>
> We list this after the first Lemma 1 statement: “Due to space constraints, all the proofs are provided in the Appendix” (line L207).
>
> > question-“i. What about the efficiency of the proposed algorithm?”
>
> Our paper’s goal is to i) develop a theoretical characterization of multi-domain Markov property, and ii) develop a sound learning algorithm that can discover the corresponding Markov EC from data. Before an efficient algorithm for learning can be developed, it’s vital to first provide a theoretical characterization of the ancestral graph and demonstrate an instance of the algorithm, confirming that such learning is realizable. A promising direction for future research would be to develop an even more efficient algorithm.
>
> After contextualizing our approach, we acknowledge that this is a practical concern. The runtime complexity of the proposed algorithm mirrors that of the standard FCI, but with the added complexity of comparing different distributions to learn the edge-structure of the augmented nodes. Various methods were proposed to alleviate this issue within the FCI framework. For instance, the RFCI algorithm [Colombo et al. 2012] attempts to construct a “possibly d-separating set” between every two nodes to select the conditioning set. In fact, this extension is discussed in the original submission’s Appendix **“Section D.1.6 Results improving efficiency of skeleton discovery phase”**. We showed that the RFCI strategy can be extended and also applies to improving S-FCI’s complexity.
>
>  > q-“ii) Are the assumptions on the multi-domain setting too strong? How could we relax them?”
>
> **Re soft intervention assumption:** We assume that interventions are soft rather than hard. Currently, no characterization for a Markov EC exists in the context of hard interventions. Each hard intervention modifies the graph, complicating the establishment of a consistent object for constraints collection. Proving the analogous to Proposition 1 (Graphical S-Markov Property) in this context is highly non-trivial due to intervention-induced graph changes.
>
> For clarity, we highlight two points.
> 1. we do not address the characterization of equivalence classes stemming from hard interventions, including those in ‘basic’ single-env.
> 2. our work sheds light on properties and nuances in hopes of paving the way for a more expansive characterization in the future.
>
> **Re shared causal structure assumption**: As noted above, managing a “changing structure” is already challenging even within a single domain. Once the skeleton starts changing arbitrarily across domains, the task of identifying a compact representation for the equivalence class becomes even more difficult. We consider this an intriguing direction and reserve its exploration for future work.

---

> > ### Comment · Reviewer_9ikg · 2023-08-14
> > **Response to the author feedback**
> >
> > After reading the author's feedback and the other reviewer's comment. I tend to agree with WVJw that the motivation and evaluation methods are not solid enough.

---

> > > ### Author Response · Authors · 2023-08-19
> > >
> > > Dear reviewer 9ikg,
> > >
> > > In light of the reviewer WVJw’s response and our clarifications (link: https://openreview.net/forum?id=C9wTM5xyw2&noteId=JsalrU9qoY). We wanted to respectfully ask if there is anything specific you would like us to address?

---

> > > > ### Comment · Reviewer_9ikg · 2023-08-19
> > > > **Further responses**
> > > >
> > > > Thanks for your efforts in the clarification. Some of my concerns have been addressed. I will increase my score to 5. Although this is a positive score, I will lower my confidence, since many parts of the paper are hard for me to fully understand due to the complex notations and representations.

---

### Official Review · Reviewer_KPtn · 2023-07-03

**Soundness:** 3 good
**Presentation:** 3 good
**Contribution:** 3 good
**Rating:** 6
**Confidence:** 3

**Summary:**

This paper develops a structure learning algorithm that uses observational and experimental data across multiple domains while handling both known and unknown target interventions. The developed algorithm generalizes and subsumes pre-existing methods that only work for single domain settings. Some experiments are given in the appendix.

**Strengths:**

The paper provides many examples when introducing notation and concepts. There is also discussion contrasting the contributions of this paper against prior works in the appendix.


**Weaknesses:**

The notation is heavy and there are quite a few typos/inconsistent notation which made the initial reading confusing; but I believe these can be remedied. See "Questions" section.

I did not check all the proofs in detail, but I do not see any glaring weaknesses.

**Questions:**

Line 2:
Do you mean "Commonly, one even collects data across multiple domains..." or "Commonly, one collects data across even multiple domains..."?

Line 18; Theorem 3:
The proposed algorithm is sound. Is it also complete? Could there be cases where it fails? Please discuss.

Suggestion to go along with Table 1:
It would be *very* helpful to include definitions of Markov, I-Markov and $\Psi$-Markov in this section, and discuss/compare them. This will allow readers to more easily follow and appreciate the subsequent discussions.

Line 36:
Is L latent while X and Y are observed? It was not clear, but I assume so.

Line 84:
What happens if we use hard interventions? Do things break?

Line 85:
Repeated "= $\langle P^1, \ldots, P^N \rangle$"

Definition 2.2:
A picture would be helpful.

Known intervention indices $\mathcal{K}$ in Definition 2.3:
$\mathcal{K}$ is defined on Line 159 and only appears later on Line 168, without appearing anywhere in the definition. Is there a purpose in explicitly mentioning it? What am I missing?

Line 163:
$X,Y,W$ are disjoint. What about $Z$ and $K$?

Line 166:
Why is $\Psi^j_\ell \in \Psi^i$? Are you saying $i = j$? If so, why introduce $j$? Or do you mean $\Psi^i_k, \Psi^j_\ell \in \Psi^\pi$?

Suggestion for Line 168:
Maybe directly write $W \setminus K$ and $K \setminus W$? That way, you can avoid introducing extra notation such as $W_K$. Also, is there some formatting issue? Definitions of $W_k$, $R$ and $R(W)$ should be part of point 2, right?

Condition 2 of Definition 2.3:
From my understanding, the intuition of (ii) is that "Y can only be affected by $S^{i,j}$ through $W$". How should I intuitively explain (i) to someone else? What is the correct intuition to have in mind?

Example 1 and Figure 1:
- Where is $S^{1,2}_x$ in the picture?
- Is there a reason why $F_{U_x}$ and $F_x$ are drawn differently?

Line 177:
"Define ... be ..." should be "Let ... be ..." or "Define ... as ..."

Line 184:
What do you mean by "constraints of type 2 are not applicable"?

Line 185:
What is $D_{\underbar{X}}$?

Typo on Line 212:
"... implies that ..." refers to condition 1 right? In that case, shouldn't it be "$P^i_j(Z \mid Y, X) = P^i_k(Z \mid Y)$" instead?

Typo on Line 223:
"$\{Y\}^1$" should be "$\{X\}^1$", right?

Suggestion for Line 228:
Add "we have (" after the first comma?

Line 241:
There is no $S_x$ in Figure 1 though?

Lines 243 to 245:
This distinction should be emphasized earlier. In my head, I was treating mechanism shift and (soft) interventions as synonyms, and I was confused by Figure 1. I suggest to put this distinction note in the preliminaries, before Section 2.

F-nodes:
If I am not mistaken, the first instance of F-nodes appear on Line 245 (and many times following that line). What are F-nodes? Section 1 only mentions S-nodes. F-nodes ned to be properly defined and discussed earlier before using them in this section.

Definition 2.4:
A picture would be helpful.

Confusion about notation of $F$ in Definition 2.4 and subsequent usage:
On Line 272, is $\binom{(}{N},2$ supposed to be $\binom{N}{2}$? Is the superscript of $F$ supposed to be single-valued and pair-valued? Subsequent discussion seem to toggle between them and also use $j \in [N]$ instead of $j \in \binom{N}{2}$. I was very confused by this; my best guess is that $F^{i}$ is shorthand for $F^{i,i}$. Is that true? Please make this clear. Thanks!

Line 273:
The symbol 'k' already means something in this context. Maybe use $\ast$ or $z$ or some other symbol for the subscript?

Proposition 1:
- Where is $F_{\mathcal{E}}$ used in Proposition 1?
- Line 285, Equation (1): What is $S_i$? Do you mean $S^i$? But I thought the superscripts of S-nodes are in pairs?
- Line 285, Equation (1): What is $F_{[k]}$? Do you mean $F_1 \cup \ldots \cup F_k$ or $F_{\mathcal{E}}$ or something else?

Typo on Line 291:
Missing "(" between "by" and $X$. I also suggest adding a comma before "and".

Typo on Line 311:
Missing ")" at the end of definitions of both $M_1$ and $M_2$.

Typo on Line 314:
Extra space: "i s".

Grammar on Line 357:
"known-target node $Y$" reads nicer than "node $Y$ known-target"

Line 357:
What is $H^{i,j}_k$? I see it appearing in Algorithm 1 but I don't think it has been properly defined.

Line 613:
Missing ")" after "given $Z$"

**Limitations:**

Nil.

---

> ### Author Rebuttal · Authors · 2023-08-10
>
> > Line 2, 36, 85, 163, 166, 168, 177, 212, 223, 228, 243 to 245, 273, 291, 311, 314, 357, 613,
>
> Thank you. We have fixed these typos, or added your suggestion into the text.
>
> > Line 18; Theorem 3: Is it also complete? …
>
> The proposed algorithm generalizes IFCI for the orientation of inducing paths. However, it’s worth noting that IFCI R9 itself isn't complete in the known-targets setting. As an illustration, consider Figure S2(a) (attached figure PDF) with interventions: {}, ${X, Y}$ and $\mathcal{K} = [1, 1]$, applying IFCI in single-domain yields PAG in S2(b). Still, we could infer $Y \rightarrow Z$ due to an inducing path. The current orientation rule with known-targets is therefore sound but incomplete. We’ve included this example in the Appendix to motivate future work.
>
> Additionally, it’s worth noting that it took over a decade to fully characterize FCI (1990s to Jiji Zhang’s 2008 work). Thus, we hope our initial characterization is a valuable step towards further investigation and understanding.
>
> > Suggestion to go along with Table 1…
>
> Due to space constraints, incorporating this into the main text would be challenging. Still, we do agree that a discussion and comparison would be useful, so we have added a section in the Appendix comparing the diff. markov properties.
>
> > Line 84: What happens if we use hard interventions?...
>
> No characterization for a Markov equiv. class w.r.t. hard interventions exists. Each hard intervention modifies the graph, complicating the establishment of a consistent object for constraints collection. Proving an analogous Proposition 1 (Graphical S-Markov Property) in this context is highly non-trivial due to intervention-induced graph changes. As a result, the process of mapping invariances to the d-separation of an augmented graph is difficult.
>
> For clarity, we highlight two points. First, we do not address the characterization of equivalence classes stemming from hard interventions, including those in ‘basic’ single-env. Second, our work seems to shed light on properties and nuances in hopes of paving the way for a more expansive characterization in the future.
>
> > Definition 2.2: A picture would be helpful.
>
> Figure S6 in the original submission contains a figure showing the depiction of this “joint selection diagram”. As this def. is not absolutely necessary, we moved this to the Appendix to conserve space for a better introduction of notation.
>
> > Known intervention indices $\mathcal{K}$ in Definition 2.3 … Is there a purpose in explicitly mentioning it? What am I missing?
>
> It is necessary because in the S-Markov Characterization (Thm 2; L306), for two tuples of $\langle G, \mathbf{\Psi}, \mathbf{S} \rangle$ to be S-Markov equivalent, they must have the same known-targets by definition. It would not make sense to compare tuples where the known-targets become unknown and vice-versa. Indeed, this is not explicitly stated, so as part of the notation fix, we have added it explicitly to Thm 2 (pg 7).
>
> > Condition 2 of Definition 2.3: How should I intuitively explain (i) to someone else?...
>
> In cond. 2, part i), it looks at whether or not the distribution of a variable y given some variable w is the same across different domains and intervention target sets. To determine this cross-distribution invariance, we look for d-separation in a surgically altered graph, where the alterations are made based on the intervened nodes. This entire def. then maps to a d-separation statement in the augmented graph in the case of structure-preserving soft-interventions in Proposition 1 (L280).
>
> > Example 1 and Figure 1: Where is $S^{1,2}_x$  in the picture?
>
> $S^{1,2}_x$ is not shown in the Figure. Example 1 (L176-180) specifies a setting using Figure 1 and adds a S-node, $S^{1,2}_x$. We have clarified this in the text.
>
> > Line 184: What do you mean by "constraints of type 2 are not applicable"?
>
> In this ex., we mean that since there is no d-separation in the surgically altered graph, then the conditional invariance in the S-Markov property (L166-167) is not required.
>
> > Line 185: What is $D_{\underline{X}}$?
>
> In preliminaries L78-79, this mean “all edges outgoing from X are removed”.
>
> > Line 241: There is no $S_x$ in Figure 1 though?
>
> $S_x$ is meant to be a S-node that points to X in the figure (Preliminaries L124). We have adapted the text to describe this more clearly.
>
> > F-nodes: What are F-nodes? Section 1 only mentions S-nodes. F-nodes ned to be properly defined and discussed earlier before using them in this section.
>
> See L238. F-nodes are auxiliary nodes representing invariances between different distributions introduced in [Pearl 1993, Dawid (2002), Jaber 2020]. These graphically encode the changes due to an intervention and enable the distributional invariances to be encoded via d-separation. We have added discussion about this to Section 1.
>
> > Definition 2.4: A picture would be helpful.
>
> Figure 2a in the main submission (pg 8) contains an example. We adjusted the text to elaborate on Def 2.4.
>
> > Is the superscript of F supposed to be single-valued and pair-valued?...
>
> Yes, that is correct. L273-274 clarify this as $F_k^i$ represents pairs of distributions within domain i and $F_k^{i,j}$ as pairs between domain i and j. We will add this to Section 1 to improve clarity.
>
> > Line 285, Equation (1): What is $S_i$?...
>
> Yes, thank you for catching this. Indeed, it should say $S^{i,j}$.
>
> > Where is $F_{\epsilon}$ used in Proposition 1? Line 285, Equation (1): What is $F_{[k]}$?
>
> Thanks for catching this! Indeed, $F_{[k]}$ should be $F_{\epsilon}$, the set of all the F-nodes added in the augmented graph. These F-nodes capture the invariances when comparing every pair of distributions included across domains.
>
> > Line 357: What is $H_k^{i,j}$?...
>
> L344: $H_k^{i,j}$ is the set of known-intervention targets; it keeps track of the symmetric differences $\mathbf{\Psi}^i_k \Delta \mathbf{\Psi}^j_l$. We have added this in the header of the Algorithm 1.

---

> > ### Comment · Reviewer_KPtn · 2023-08-10
> >
> > Thank you for the detailed response. I appreciate the effort the authors have taken into this.
> >
> > While I agree with the other reviewers that this paper is *very heavy* on notation and can be hard to follow, I can feel the authors' efforts to make them easy to understand via various diagrams. Furthermore, the authors have also promised to improve their writing in the revision (which I trust they will). I **strongly disagree** with the other reviewers that this paper is poorly motivated and also believe that this is a non-trivial theoretical contribution to the field. In particular, I find that it is unfair that the paper has been judged poorly because it lacked strong empirical evaluations --- the NeurIPS Call For Papers (CFP) welcomes theoretical submissions and I have seen many good theoretical works published in past NeurIPS-es. I am willing to defend my score and I urge the other reviewers to appreciate the paper from a more theoretical perspective.
> >
> > Two follow-up questions:
> > - I think there is a typo in the attached PDF: the caption of Figure S2 should be saying "not complete" instead of "not sound"?
> > - I was under the impression that hard interventions can be viewed as special cases of soft interventions (just ignore any incoming causal influence from parents), so I always thought that any process that can handle soft interventions would also be able to handle hard interventions (for free, essentially). As such, I'm still not really understanding your response for Line 84. If it does not take too much of your time, could you kindly clear my misunderstanding on this matter? Thank you very much.

---

> > > ### Author Response · Authors · 2023-08-11
> > > **Response to Figure S2, and hard vs soft interventions in structure learning setting**
> > >
> > > Thank you very much for your detailed feedback and support of our paper. Please, see answers below.
> > >
> > > > I think there is a typo in the attached PDF: the caption of Figure S2 should be saying "not complete" instead of "not sound"?
> > >
> > > Yes, indeed this is true. Thank you for catching this glitch! The correct caption should state: “Counter-example demonstrating that orientation rules for known-targets stemming from I-FCI are not complete.”
> > >
> > > > I was under the impression that hard interventions can be viewed as special cases of soft interventions (just ignore any incoming causal influence from parents), so I always thought that any process that can handle soft interventions would also be able to handle hard interventions (for free, essentially). As such, I'm still not really understanding your response for Line 84. If it does not take too much of your time, could you kindly clear my misunderstanding on this matter? Thank you very much.
> > >
> > > The literature is somewhat inconsistent in this matter, unfortunately, so we appreciate the opportunity to clarify this important issue.  In the context of effect identification, hard interventions are usually cast as a special case of soft interventions. For example, [Correa et. al. 2020] uses soft (or stochastic) to refer to an intervention that replaces $P(X|Pa_x,U_x)$ with $P^*(X|Pa^*_x)$ where $Pa^*_x$ and $Pa_x$ are not necessarily the same. Hence, it could lead to structural changes in the graph.
> > >
> > > On the other hand, a soft intervention in the structural learning context in our paper (and related works, [Eberhardt 2005, Yang 2019, Jaber 2020, Kocaoglu 2019]) replaces $P(X|Pa_x,U_x)$ with $P^*(X|Pa_x,U_x)$ such that $P(X|Pa_x,U_x) \neq P^*(X|Pa_x,U_x)$; it assumes the parents remain the same but the distribution is altered. This does not lead to structural changes in the graph.
> > >
> > > To understand the challenges with hard interventions as opposed to soft, consider the following example graphs:
> > >
> > > $G1 = (1 \leftrightarrow 2 \leftrightarrow 3 \leftrightarrow 4, 2 \rightarrow 4, 3 \rightarrow 1)$ and
> > > $G2 = (1 \leftrightarrow 2 \leftrightarrow 3 \leftrightarrow 4, 2 \rightarrow 4, 3 \rightarrow 1; 1 \leftrightarrow 4 )$
> > >
> > > The only structural difference is that G2 has an extra bidirected edge between ‘1’ and ‘4’. Under soft interventions on ‘2’ , one cannot distinguish them apart due to the inducing paths. However, assuming the intervention is hard, we can learn that ‘1’ and ‘4’ are not adjacent in G1 by breaking the inducing path between them. Hence the two graphs are distinguishable under the hard intervention but not the soft intervention. The technical challenge is that the I-MAG representation (and its framework), first suggested in [Kocaoglu 2019] and utilized in our work, is not sufficient to capture these subtleties.
> > >
> > > # References
> > >
> > > [1] Correa, Juan, and Bareinboim, Elias. "A calculus for stochastic interventions: Causal effect identification and surrogate experiments." Proceedings of the AAAI conference on artificial intelligence. Vol. 34. No. 06. 2020.
> > >
> > > [2] Kocaoglu, Murat, et al. "Characterization and learning of causal graphs with latent variables from soft interventions." Advances in Neural Information Processing Systems 32 (2019).
> > >
> > > [3] Jaber, A., Kocaoglu, M., Shanmugam, K., & Bareinboim, E. (2020). Causal discovery from soft interventions with unknown targets: Characterization and learning. Advances in neural information processing systems, 33, 9551-9561.
> > >
> > > [4] Yang et . al. 2019. "Characterizing and Learning Equivalence Classes of Causal DAGs under Interventions"
> > >
> > > [5] Eberhardt, F., Glymour, C., and Scheines, R. On the number of experiments sufficient and in the worst case necessary to identify all causal relations among n variables. In Uncertainty in Artificial Intelligence, pp. 178–184, 2005.

---

> > > > ### Comment · Reviewer_KPtn · 2023-08-12
> > > >
> > > > Thank you very much for the clarification and references!

---

> > > > > ### Author Response · Authors · 2023-08-17
> > > > >
> > > > > Thank you very much for the engaging discussion and thoughtful questions. We just wanted to respectfully follow-up to check if there were any other clarifications we can provide? We hope the understanding from this discussion could be reflected in our score.

---

> > > > > > ### Comment · Reviewer_KPtn · 2023-08-17
> > > > > >
> > > > > > No worries, I'm sorry that I did not make it explicit earlier: I am intending to maintain my (positive) score.

---

### Official Review · Reviewer_WVJw · 2023-07-11

**Soundness:** 2 fair
**Presentation:** 1 poor
**Contribution:** 3 good
**Rating:** 4
**Confidence:** 3

**Summary:**

This paper analyzes the categorization and properties of Markov equivalence class from interventional data. The unique part is that this paper considers multiple environments (use env for short lately), where in each env, both observational and interventional data are accessed.

**Strengths:**

1. Causal discovery from interventional data is a promising direction, because genuine causal relations can be recovered by RCT, where RCT generates interventional data.
2. The setting is general, which considers the existence of latent confounders, and also extend the previous setting to {obs, intv} across different envs.

**Weaknesses:**

1. The motivation is not convinced.
first of all, the term "env" here is different from conventional usage. below [1][2][3] shows the conventional meaning of the term "environment", which means that an env just alters the observational distribution by known or unknown underlying mechanism. Based on my understanding, in the standard definition of "causal discovery from interventional data", each interventional data (a sample from the causal graph under known/unknown interventions) is viewed as collected from an env.
I think the above setting is general enough to meet real needs, what is the motivation to build a second layer of "env" on top of that? If this is indeed the case, I believe it should be supported by solid empirical evaluations.
[1] Huang, B., Zhang, K., Gong, M., & Glymour, C. (2019, May). Causal discovery and forecasting in nonstationary environments with state-space models. In International conference on machine learning (pp. 2901-2910). PMLR.
[2] Peters, J., Buhlmann, P., & Meinshausen, N. (2015). Causal inference using invariant prediction: identification and confidence intervals. arXiv. Methodology.
[3] Peters, J., Janzing, D., & Schölkopf, B. (2017). Elements of causal inference: foundations and learning algorithms (p. 288). The MIT Press. (Section 7.2.5)

2. lacks solid empirical evaluations to support the claims
the second contribution of this paper is that "Learning algorithm - We develop a sound learning algorithm for learning a Markov
equivalence class of selection diagrams with observational and/or interventional data", so I wonder
a) any empirical justification about the soundness?
b) compared with existing work, what advantage of the setting, i.e., {obs, intv} across envs?
c) any real-world usage to support the motivation, or to convince readers that the categorization and analysis in this paper are indeed needed.
Currently, there is no experiment part in main body, this is unacceptable from my opinion.

3. The notation part is extremely hard to follow.

**Questions:**

The paper mentioned in introduction about the work of JCI [1], and argues that JCI pools data together and perform learning
on the combined dataset [35]. Pooling data is an incorrect procedure, ... may result in learning an invalid MEC (see Example 11).
So this paper argues that even under the standard setting (unknown interventional targets from observational and interventional data, under a single env), JCI is incorrect. I checked example 11, but it links to example 6 and other properties. So can you elaborate one such example to point to the mistakes made by JCI?

[1] Mooij, J. M., Magliacane, S., & Claassen, T. (2020). Joint causal inference from multiple contexts. The Journal of Machine Learning Research, 21(1), 3919-4026.

---

> ### Author Rebuttal · Authors · 2023-08-10
>
> > w - “1. The motivation is not convinced… the term "env" here is different from conventional usage… that an env just alters the observational distribution by known or unknown underlying mechanism…. what is the motivation to build a second layer of "env" on top of that?”
>
> Thank you for the opportunity to elaborate on this important issue. The question entailed by your note is: How do interventions and environments differ in real-life scenarios?
>
> Many examples across scientific disciplines demonstrate that the notions of domain/environment and interventions are distinct. For example, when making inferences about humans based on data from bonobos, this distinction becomes clear. The difference between the two species is depicted as the environment/domain in this context. A scientist might perform an intervention on a bonobo's kidney (specifically, what we're representing as Z), and try to determine the effect of medication (X) on fluid equilibrium in the body (Y). Although we could intervene on Z in bonobos and observe its effect on X and Y, our ultimate goal might be to understand the effect of X on Y in humans. It's generally invalid to conflate these two qualitatively different indices, a point first noted by Bareinboim & Pearl in the context of transportability analysis around 2011. The distinct environments exist regardless of any intervention, such as medication. Also, an intervention on the kidney function is different across the two species. Bareinboim & Pearl (2011) formalized this setting, introducing clear semantics for the S-nodes (environments) that essentially offers a combined representation for both environments. We appreciate the references listed though and have added them to the introduction. With this foundation, we can now address the reverse problem of determining the equivalence class of selection diagrams from data.
>
> Moving to mathematics, neglecting these distinctions has significant implications for the structure learning problem. To witness, consider Figure 2 in the main text. It demonstrates, even in a simple 2-node scenario, why environments and experiments must be distinguished. We have a laboratory (domain 1) and a hospital setting (domain 2). In the lab, we can experimentally perturb proteins (Y) and simultaneously observe levels of biomarker (X). Let the true causal direction be $X \rightarrow Y$.  In the hospital, observational data is collected, which includes sequencing data and biomarker values. Given a domain-shift (e.g. batch effects common in gene sequencing exps), it’s essential to account for the shift when comparing observational and interventional data. Otherwise, one might mistakenly infer that $Y \rightarrow X$. While this orientation is valid if the data were collected in a single-domain setting and the same distributional invariances held, it isn’t accurate in this specific example due to the overlooked domain shift. In other words, a single indicator can’t describe the invariance in this setting, equating interventions and domain-shifts is erroneous.
>
> We further demonstrate differences wrt JCI [Mooij 2020] and ICP [Peters 2015] in a few simple examples shown in Figure S4 and S5 (attached fig. PDF).
>
> The framework of “nonstationary changes” [Huang 2019] also uses auxiliary random variables to capture mechanism changes as a result of time-series non-stationarity and thus can be seen as encompassed within the JCI framework. They allow changes in causal strengths and noise variances though, which may be interesting to explore in future work and how it can improve the characterizations and/or learning we propose here.
>
> > w: “2b) compared with existing work, what advantage of the setting, i.e., {obs, intv} across envs?”
>
> We have shown in the Appendix “Section D.2 Comparisons with Other Works” (pg 25)  how the S-Markov property and the SFCI algorithm generalize the existing $\Psi$-Markov and $\Psi$-FCI algorithm.
>
> > w: “2a) any empirical justification about the soundness?”
>
> Due to space constraints, we put empirical exps. in the Appendix Section D.3 (pg 26). We ran SFCI on the Sachs dataset shown as Figure S5 in the submission.  We also added Figure S10 (attached fig. pdf), which demonstrates in a simulation how S-FCI compares with $\Psi$-FCI and standard FCI algorithm.
>
> > w: “2c) any real-world usage to support the motivation,...
>
> Along with the previous response, in Table 1 we highlight different bodies of scientific work that look at different domains with obs. and/or interventional data. Furthermore, we ran S-FCI on the Sachs dataset in Appendix Sec. D.3.1 (pg 27).
>
> > w: “3. The notation part is extremely hard to follow.”
>
> We understand the notation is complex due to carrying around the domains, distributions and intervention targets. We have modified the paper and introduced objects and notation in a more organized format in the Preliminaries shown in the author rebuttal. We hope we have motivated the need for notation that distinguishes domains, interventions and distributions.
>
> > q: “The paper … argues that JCI pools data together and perform learning on the combined dataset. Pooling data is an incorrect procedure, ... may result in learning an invalid MEC (see Example 11)...”
>
> Thank you for pointing this out!. Our choice of wording has been changed to **“Pooling data is an incomplete procedure… and moreover ignoring the presence of a domain-shift with interventional data can lead to learning an invalid MEC (see Example 11)”**.
>
> **Re incompleteness of the JCI procedure [2]:** In the Appendix of [Jaber et al. 2020], Section D.2 Prop. 6, shows that, if there are at least 3 distributions, the JCI procedure does not capture all the invariances across every pair of distributions required by the $\Psi$-Markov property. The same reasoning applies here since the S-Markov property generalizes the $\Psi$-Markov property.  The incompleteness of the pooling procedure is also shown in an example comparing JCI to SFCI in Figure S5 (fig. rebuttal).

---

> > ### Comment · Reviewer_WVJw · 2023-08-15
> >
> > Thank you for your reply. The motivation of introducing the concept of domain variable, and the distinction between domain and intervention is much clearer. As an audience from the area of "causal discovery from interventional data", I suggest the authors to significantly improve the motivation part, to better convey the key concept of domain variable and its distinction with interventional variables.
> >
> > One follow up question, based on my understanding of the S-nodes, which specifies the different domains, I view each S-node can be viewed as a soft intervention, with known target. Therefore, I want to see that there is no distinction between S-node and interventional nodes in general, thus hoping the problem can be reduced to a certain setting under standard "causal discovery from interventional data". Take the lab and hospital example you provided, I can provide an augmented data with 4 variables {X, Y, I_lab, I_hos}, where I_lab and I_hos are augmented variables. Let's assume no latent confounders, then the augmented graph has the following edges: I_lab ->X (indicates a domain distribution to generate X in lab), I_lab->Y (indicates experimentally perturb proteins Y); I_hos->X (indicates another domain distribution to generate X in hos). X-Y (indicates there is a dependency between X and Y, but the direction is unknown). Now, can the problem be re-casted as, given such augmented data (by pooling the data together just as what JCI did), and given such augmented graph, can we infer the direction between X and Y?

---

> > > ### Author Response · Authors · 2023-08-16
> > >
> > > > Thank you for your reply. The motivation of introducing the concept of domain variable, and the distinction between domain and intervention is much clearer. As an audience from the area of "causal discovery from interventional data", I suggest the authors to significantly improve the motivation part, to better convey the key concept of domain variable and its distinction with interventional variables.
> > >
> > > Thank you for reading through our response. We have taken your feedback into consideration and will more explicitly convey the distinction between domain-changes and interventions in the introduction for an improved explanation of the motivation.
> > > > I view each S-node can be viewed as a soft intervention, with known target…
> > >
> > > Just to clarify, each S-node actually can be viewed as a soft intervention with **unknown** target, not known. This is because you do not know which variables’ mechanisms are altered.
> > > It is worth mentioning that known and unknown targets are important to distinguish for both characterization and learning. For example, if we have $G = (X \rightarrow Y; X \leftrightarrow Y)$ and an intervention on X. If the intervention is known-target, then every causal graph in the I-Markov Equivalence Class (EC) contains the edge $X \rightarrow Y$ [Kocaogulo 2019]. However, if the intervention is unknown-target, then both $X \rightarrow Y$, or $X \leftarrow Y$ are possible edges in some causal graph in the corresponding $\Psi$-Markov EC [Jaber 2020].
> > >
> > > > Therefore, I want to see that there is no distinction between S-node and interventional nodes in general, thus hoping the problem can be reduced to a certain setting under standard "causal discovery from interventional data".
> > >
> > > Recall from our previous comment that a domain change (denoted by an S-node) can be perceived as an intervention with **unknown** targets. At the same time, we could still have interventions with **known** or **unknown** targets in each domain. In light of this, simply applying the machinery for unknown targets yields an incomplete characterization of the equivalence class while applying the theory for known targets yields an incorrect characterization. This motivates the need for a more general framework that can factor in both settings, which is what we propose in this work.
> > > > Take the lab and hospital example you provided, I can provide an augmented data with 4 variables {X, Y, I_lab, I_hos}, where I_lab and I_hos are augmented variables. Let's assume no latent confounders, then the augmented graph has the following edges: I_lab ->X (indicates a domain distribution to generate X in lab), I_lab->Y (indicates experimentally perturb proteins Y); I_hos->X (indicates another domain distribution to generate X in hos). X-Y (indicates there is a dependency between X and Y, but the direction is unknown). Now, can the problem be re-casted as, given such augmented data (by pooling the data together just as what JCI did), and given such augmented graph, can we infer the direction between X and Y?
> > >
> > > We hope to have understood your question, but **please let us know otherwise** or if something else is unclear and deserves further elaboration.
> > >
> > > To answer your question, we start with the observation that [Squires et al. 2020; Jaber et al. 2020] showed, namely, that interventional Markov EC with known and unknown-targets are the same in the setting of causal sufficiency (i.e. no latent confounders). However, considering the more general setting that we consider in our paper, this is no longer true when unobserved confounders (UC) are present in the system. Thus, adding UC to the example will help illustrate the subtlety.
> > >
> > > To expand on your example and considering the presence of UCs, we add a latent confounder between X and Y. The corresponding graph is $G = (X \rightarrow Y; X \leftrightarrow Y)$, building on the example raised above in discussion on S-nodes. Assume that the mechanism of X is different between the two domains (lab and hospital). Thus, we have an S-node pointing to X. Also, assume there is an observational dataset in the hospital, an observational dataset in the lab, and an interventional dataset (where X is randomized) in the lab. The corresponding augmented graph is $Aug(G) = (X \rightarrow Y; X \leftrightarrow Y; F_x \rightarrow X; S^{lab, hos} \rightarrow X)$.
> > >
> > > Based on the discussion earlier, if we only use the theory for interventions with unknown-targets (i.e., treat F_x and S as F-nodes for unknown targets), both $(X \rightarrow Y; X \leftrightarrow Y)$ and $(X \leftarrow Y)$ are part of the interventional Markov EC. However, we can use the known intervention on X (denoted by F_x) to rule out $(X \leftarrow Y)$ from the EC. On the other hand, as illustrated earlier, each S-node actually can be viewed as a soft intervention with **unknown** target, not known. In summary, and going back to your question, JCI does not learn that X is a cause of Y in this setting, while the newly proposed treatment is capable of doing so.

---

> > > > ### Author Response · Authors · 2023-08-18
> > > >
> > > > Thank you for the engaging discussion.
> > > >
> > > > Since the reviewing period is ending soon, we just wanted to respectfully follow up with the reviewer to see if there are any additional clarifications we may provide? We believe we have resolved all the issues raised, but are happy to discuss more.

---

> > > > > ### Comment · Reviewer_WVJw · 2023-08-20
> > > > >
> > > > > Thanks for your clarification. In the mixture of both known and unknown interventional targets (the former mainly is from RCT, and the latter can be from domain shift), and in the presence of latent confounders, the motivation of the work becomes valid.
> > > > >
> > > > > Overall, I suggest a major revision of this paper from two perspectives:
> > > > > 1) The motivation of introducing the concept of domain variable, and the distinction between domain and intervention.
> > > > > 2) a better illustration of the problem formulation, especially the relationship with existing works on modeling known/unknown interventions, in the setup of whether latent confounders are considered or not.
> > > > > Once these are addressed, I believe this paper will be more appreciated and have better impact to the community.
> > > > >
> > > > > I slightly updated my review status, but overall, regarding the current paper body, I'm still concerned of its readiness for a publication, I believe the introduction and related work should be carefully prepared, and I suggest a significant top-down organization of the tech body to convey the key ideas, rather than showing too many details in very early stage. Thanks!

---

> > > > > > ### Author Response · Authors · 2023-08-21
> > > > > >
> > > > > > Thank you for your response and review over this period.
> > > > > >
> > > > > > > 1. The motivation of introducing the concept of domain variable, and the distinction between domain and intervention.
> > > > > >
> > > > > > In summary, we have incorporated the following changes as a result of this review:
> > > > > >
> > > > > > The discussion presented in https://openreview.net/forum?id=C9wTM5xyw2&noteId=lJ7gwKh2Po is now in our paper’s Introduction section now to better motivate the distinction between interventions and domain-changes.
> > > > > > We have added Figure S4 and S5 to the Appendix along with a detailed discussion comparing our work to ICP and JCI.
> > > > > >
> > > > > > We believe this addresses the issue, but please let us know if this is not the case.
> > > > > >
> > > > > > > 2. a better illustration of the problem formulation, especially the relationship with existing works on modeling known/unknown interventions, in the setup of whether latent confounders are considered or not.
> > > > > >
> > > > > > Based on our discussion in the previous thread: https://openreview.net/forum?id=C9wTM5xyw2&noteId=vZEyzCkBVm, we will reflect this discussion in the Introduction Section of our paper with comparisons with existing work on causal discovery using unknown and known target interventions in the single domain setting. We hope this addresses the issue for the reviewer.

---

### Author Rebuttal · Authors · 2023-08-10

We thank the reviewers for their time and effort reading our paper and for your valuable feedback with this review. We have attempted to address each of your raised points and hope you can consider the paper in the light of the clarifications provided in your respective rebuttals.

Overall, we have included a better formatted section in the preliminaries to describe the notation, which provides a more structured introduction to the distinct, but necessary objects in our paper.

**[Notation]** Since, domains, interventions, and distributions are central distinctions throughout the whole paper, we added the following to “Section 1.1 Preliminaries and Notation” to simplify the introduction of this notation.

 “

The following objects are utilized repeatedly, and introduced here. Our notation borrows from the transportability literature [Bareinboim & Pearl, 2012].

1. **Domains**: $\mathbf{\Pi} = \{ \Pi^1, \Pi^2, ..., \Pi^N\}$ denotes a set of N domains.
2. **Intervention targets**: $\mathbf{\Psi}^\Pi = \langle \Psi^1_1, \Psi^1_2, ..., \Psi^N_M \rangle$ is an ordered tuple of sets of intervention targets, with different sets of intervention targets occurring within each of the N domains for a total of M intervention target sets. We will denote $\mathbf{\Psi}^i$ as the intervention targets associated with domain i.
3. **Distributions**: $\mathbf{P}^\Pi = \langle P^1_1, P^1_2 ..., P^N_M \rangle$ is an ordered tuple of probability distributions that are available to learn from. Denote $\mathbf{P}^i$ as the distributions associated with domain i. There is a one-to-one correspondence between $\mathbf{P}$ and $\mathbf{\Psi}$, such that $P^i_j$ is the distribution associated with targets $\mathbf{\Psi}^i_j$ in domain i.
4. **Known target indices**: $\mathcal{K}$ is a vector of 1’s and 0’s indicating, which sets of interventions are known-targets. $\mathcal{U} := 1 - \mathcal{K}$ represents therefore an index vector selecting the distributions and interventions with unknown targets. $\mathbf{P}\_{\mathcal{K}}$ and $\mathbf{\Psi}_{\mathcal{K}}$ denotes the set of distributions and intervention targets corresponding to the known target interventions.
5. **Causal diagram**: $G =  (\mathbf{V} \cup \mathbf{L}, \mathbf{E})$, is a shared diagram over the N domains.
6. **Selection diagram**: $G_S =  (\mathbf{V} \cup \mathbf{L} \cup \mathbf{S}, \mathbf{E} \cup \mathbf{E_S})$, extends G with the corresponding S-nodes and their edges to represent each pair of domains.

“

---

### Decision · Program_Chairs · 2023-09-21

**Decision:**

Accept (poster)

**Comment:**

The three reviewers and the authors had a robust discussion with much back and forth.  Nevertheless, in the end the reviewers could not reach consensus.  After reading it in great details myself, I agree with many of the points of all the reviewers, and the authors are strongly encouraged to take all reviewer comments into account in the final version. But I actually found the notation clear and the writing extremely clear given the challenging nature of the topic, and I found the motivation solid.  Therefore I am recommending acceptance but strongly advising the authors to address especially the third reviewer's concerns about discussion of relationship to other work (in the supplement if it won't fit into the paper) and about experiments. I think the paper makes a concrete set of useful theoretical contributions to the causal discovery field.

Other items the authors should be sure to address in the final version are:

For such a well-written paper, I was surprised “soft intervention” was used repeatedly but never defined. And it’s central to the paper! For that reason, even though I know it’s common in the literature, it should be defined again in the paper too.

Line 109: SCMs is -> SCMs are

I wasn’t sure the multiple domains problem was that common or important, but the paper gives good examples, including the very common setting of: same biological mechanism but data from different labs

Because the supplement begins by repeating the full paper, the theorem numbers in the supplement (where the proofs are) don’t seem to align easily with the theorem numbers in the paper. So it takes more work than it should for the reader (for example) to go to the supplement and find the proof of Theorem 2. At least it did for this reader.